# Sensor-Based Activity Recognition Using Frequency Band Enhancement Filters and Model Ensembles

**DOI:** 10.3390/s23031465

**Published:** 2023-01-28

**Authors:** Hyuga Tsutsumi, Kei Kondo, Koki Takenaka, Tatsuhito Hasegawa

**Affiliations:** Graduate School of Engineering, University of Fukui, Fukui 910-8507, Japan

**Keywords:** frequency emphasis, ensemble learning, deep learning

## Abstract

Deep learning methods are widely used in sensor-based activity recognition, contributing to improved recognition accuracy. Accelerometer and gyroscope data are mainly used as input to the models. Accelerometer data are sometimes converted to a frequency spectrum. However, data augmentation based on frequency characteristics has not been thoroughly investigated. This study proposes an activity recognition method that uses ensemble learning and filters that emphasize the frequency that is important for recognizing a certain activity. To realize the proposed method, we experimentally identified the important frequency of various activities by masking some frequency bands in the accelerometer data and comparing the accuracy using the masked data. To demonstrate the effectiveness of the proposed method, we compared its accuracy with and without enhancement filters during training and testing and with and without ensemble learning. The results showed that applying a frequency band enhancement filter during training and testing and ensemble learning achieved the highest recognition accuracy. In order to demonstrate the robustness of the proposed method, we used four different datasets and compared the recognition accuracy between a single model and a model using ensemble learning. As a result, in three of the four datasets, the proposed method showed the highest recognition accuracy, indicating the robustness of the proposed method.

## 1. Introduction

In recent years, the widespread use of smartphones and wearable devices has facilitated user activity sensing. These devices can perform activity recognition using accelerometer and gyroscope data as time-series data [1,2]. Activity recognition can be used, for example, to determine a user’s health status [3,4]. Activity recognition technology can also be applied to sports such as volleyball and badminton [5,6]. For enhanced service applications, it is desirable to recognize activities accurately and in detail. For this purpose, Sikder et al. [7] transformed accelerometer and gyroscope data into frequency and power spectrum and used them as input to a convolutional neural network (CNN) to classify six types of activities. That study used the frequency spectrum as input for the model and evaluate recognition accuracy but did not consider the difference in frequency characteristics between activity. Other studies have focused on the frequency characteristics of activities. Ooue et al. [8] converted accelerometer data into a power spectrum to determine the frequency characteristics of different walking patterns and found that they differed between normal walking and walking with a limp. Therefore, it is likely that the frequency characteristics of each activity will differ in activity recognition, and there may be important frequencies for the prediction of each activity. Liu et al. [9] analyzed the power spectrum of input data to obtain the major frequency bands and proposed a tree-structured wavelet neural network (T-WaveNet) for time-series signal analysis but did not perform frequency enhancement of the input data. In this study, we propose an activity recognition method that identifies the important frequency for recognizing a certain activity, applies a filter that emphasizes each frequency in the input, and performs ensemble learning during training and testing. The aim is to improve the accuracy of activity recognition and to facilitate the development of various activity recognition services using a general-purpose method based on frequency enhancement and ensemble learning. As discussed below, parts of studies on activity recognition proposed converting accelerometer data into a frequency spectrum as the input of CNN. In contrast, this study makes the following contributions:We experimentally identified the important frequency of various activities using the Human Activity Sensing Consortium (HASC) activity recognition dataset [10].We developed a new method to improve the accuracy of activity recognition by creating a filter that emphasizes the important frequency of each activity and applying it to training and testing data, training the model with the data, and using ensemble learning.

## 2. Related Research

### 2.1. Sensor-Based Activity Recognition

Various methods for sensor-based activity recognition, including CNN and ensemble learning, have been developed. Shaohua et al. [11] used three-axis smartphone accelerometer data to perform activity recognition using CNN, Long Short-Term Memory (LSTM), Bidirectional LSTM, Multilayer Perceptron, and support vector machine (SVM) models, and compared their accuracies using two large datasets. According to their experimental results, the CNN model had the highest accuracy. Ito et al. [12] performed Fourier transform processing of accelerometer and gyroscope data to create a spectrum image, which was used as input to a CNN model for activity recognition. This model had three convolutional layers and three pooling layers. After integrating the features of the spectrogram images from the accelerometer and gyroscope, classification was performed on all three fully-connected layers. The best convolution size was obtained by comparing the accuracy of different convolution sizes in the time and frequency directions. Subasi et al. [13] used ensemble learning to classify seven types of activities using random forest and SVM methods and compared their recognition accuracies with that of activity recognition using Adaptive Boosting combined with these methods. Sakorn et al. [14] used acceleration and gyro data collected by smartphones for activity recognition. They proposed a method that combines a 4-layer CNN and an LSTM network, and showed that it improves the average accuracy by up to 2.24% compared to state-of-the-art methods. Others have proposed models that combine CNNs and BiGRUs, and have shown to significantly outperform the recognition accuracy of other RNN models [15]. Nadeem et al. [16] proposed a method for extracting optimal features using sequential floating forward search (SFFS), and showed that the recognition accuracy is about 6% higher than when no features are selected. Muhammad et al. [17] proposed a two-level model and performed data recognition when multiple activities are combined. All these studies used data obtained from accelerometers and gyroscopes as input or spectrogram images to recognize activity. However, none of them used data that utilized the characteristics of each activity.

### 2.2. Frequency Characteristics in Activity Recognition

Some studies have used the frequency characteristics of activities. Yoshizawa et al. [18] used an Infinite Impulse Response (IIR) bandpass filter to detect change points from one moving activity to another. A change point was detected when the sum of the fluctuations of each component of the accelerometer data exceeded a certain value. The authors also identified the important frequency by changing the spectrum coefficients used in the change point detection method to determine the number of filters and pass frequencies of the IIR bandpass filter. Fujiwara et al. [19] applied short-term Fourier transform to Doppler sensor data to calculate the frequency components as features used to construct a lifestyle activity recognition model. To reduce the dimensionality of the feature values, they used only a portion of the frequency components. They determined the frequency components reduced by examining changes in recognition accuracy while reducing high- and low-frequency components. They found that recognition accuracy was highest when the bandwidth of the frequency components used as feature values ranged from 0 to 5 Hz. These studies have demonstrated that there is an important frequency for activity recognition. However, such frequency has been used mainly for model analysis or feature reduction and rarely for improving the accuracy of activity recognition.

### 2.3. Activity Recognition Using Ensemble Learning

Irvine et al. [20] proposed a neural network ensemble learning method for the recognition of daily activities in a smart home. Zhu et al. [21] used an ensemble learning of two CNN models to classify seven types of activities. First, they made predictions using a model that classified the seven types. Subsequently, if the results were of two specific classes, they made predictions using another model that classified these two types. They then obtained the final output by performing weighted voting on the outputs of the two models. Yiming et al. [22] proposed a method that combines extreme learning machines (ELMs) with pairwise diversity measure and glowworm swarm optimization-based selective ensemble learning (DMGSOSEN), which achieves higher recognition accuracy with fewer models than the comparison method. Other methods include a CELearning model using multiple layers of four different classifiers [23], an ensemble learning model using Adaboost and SVM [24], a model combining gated recurrent units (GRU), CNN, and deep neural networks (DNN) [25], and ensemble learning with multiple deep learning models [26]. Another study [27] applied multiple data augmentation to input data to perform activity recognition using ensemble learning but did not focus on frequency characteristics.

## 3. Proposed Method

Figure 1 shows an overview of the proposed method. The proposed method improves the accuracy of activity recognition by identifying important frequency bands for each activity, creating a filter to enhance them, and applying each technique (DA: frequency emphasis in training, TTA: frequency emphasis in testing, and EL: ensemble learning). The method consists of three phases described in Section 3.1, Section 3.2 and Section 3.3. 

### 3.1. Phase 1: Finding the Important Frequency for Each Activity

In this phase, the important frequency for each activity is obtained as follows:
The CNN model M is trained using the original accelerometer data $x_{train}$ as in general activity recognition.For acceleration data $x_{valid}$ the subjects of which differ from that of $x_{train}$, some frequencies are masked by changing *f* in Equation (1) between (0, $f_{s}$/2]:(1)$$x^{\prime }=F_{m}\left(x,f\right)=ifft\left(P\left(fft\left(x\right),f\right)\right).$$Using the model M trained in step 1, the change in the recognition accuracy of the data masked in step 2 is examined. Step 3 is performed for each activity 𝒸 to obtain the set of frequency bands to be emphasized: $ℱ=\{f^{𝒸}|𝒸\in C\}$ (Figure 1a).

Note that x∈ℝ3×w is the triaxial accelerometer data (w is the window size), fft(⋅) is the Fourier transform, ifft(⋅) is the inverse Fourier transform, P(·) is the process of masking frequency bands, Fm(x,f) is the data after mask processing, f𝒸 is the important frequency at a given activity 𝒸, and fs  is the sampling frequency of the accelerometer data. The frequency of 0 Hz is not masked because it is a DC component. The maximum frequency to be masked is fs/2 because the frequency of the Fourier-transformed data has a maximum value of 1/2 of the sampling frequency. The frequency at which the recognition accuracy decreases is considered the important frequency.

### 3.2. Phase 2: Emphasis during Training

In this phase, the CNN model $M_{c}$ is trained on the training data $x_{train}$ using ℱ calculated as described in Section 3.1, with the frequency band enhancement filter of Equation (1) applied to the data (Figure 1b). The number of models is |C| because the models are trained using data enhancing the important frequency of each activity. The frequency band weighting filter is implemented as Equation (1) where P(⋅) is the process of frequency band enhancement. The $f^{𝒸}$ obtained in Phase 1 is input to f in Equation (1).

In this study, four types of window functions were used as filters to enhance the frequency bands. Examples of the filters used are shown in Figure 2. The peak window does not change the amplitude spectrum of the important frequency of each activity as determined experimentally but multiplies the amplitude spectrum of the other frequencies by a factor of 1/2. The Gaussian window is a normally distributed window, with the important frequency of each activity as the mean and a standard deviation of 10 adjusted so that the maximum value is 1 and the minimum value is 0.5. The triangular window is a window with the amplitude spectrum of the important frequency of each activity as the vertex. The minimum value is set to 0.5. Random window is a random value of 0.5–1 applied to the (0,7.8] Hz portion of the amplitude spectrum. Using the random window, we determined whether the emphasis on the important frequency of each activity contributes to improving the accuracy of activity recognition. Figure 2 shows the filter for $f^{c}=3\;\mathrm{Hz}$.

### 3.3. Phase 3: Emphasis during Testing

In this phase, Equation (1) was applied to the testing data $x_{test}$ and inputted to the model $M_{c}$ trained in the previous phase. The final output is the result of the majority voting on the output of each model (Figure 1c). This method can be regarded as a kind of Test Time Augmentation (TTA) [28], a method in which the testing data are processed to create several types of data, where the input data are augmented by frequency band enhancement filters.

Note that in this study, in order to eliminate differences in recognition accuracy due to differences in model structure, we used VGG16 [29] as the unified model used in Phases 1, 2, and 3.

## 4. Evaluation Experiment

### 4.1. Experiment Summary

We first conducted an experiment to determine the important frequency band for each activity. We masked some frequencies in the accelerometer data and used these data as input to the model to examine changes in accuracy and identify the important frequency (i.e., the frequency at which accuracy decreased). Next, using the obtained frequency, we created a frequency band enhancement filter for each activity and applied it to the accelerometer data. We then conducted an ablation study to evaluate the contribution of the three components of the proposed method (frequency emphasis during training, frequency emphasis during testing, and ensemble learning) to recognition accuracy.

### 4.2. Experimental Setup

#### 4.2.1. Model Structure and Training Procedures

In the experiments, we used VGG16 as an activity recognition model modified for 1D data. To reduce the influence of the model’s classifier, we applied a shallow classifier using global average pooling, and the classifier was a single fully-connected layer. In training, the batch size was set to 256, the learning rate was set to 0.001, and the number of epochs was set to 200. The kernel size was set to three, the stride width was set to one, no padding, Rectified Linear Unit was the activation function, and the pooling size was set to two.

#### 4.2.2. Dataset

The HASC dataset was used for activity recognition. The sampling frequency was 100 Hz. We randomly sampled the acceleration data from 80 persons for training, 20 for validation, and another 30 for testing. The window size was 256 samples, divided into time series. Six activity labels were used: stay (standing still), walk (walking), jog (jogging), skip (skipping), stUp (climbing up a staircase), and stDown (climbing down a staircase). Accelerometer data contain noise; however, in this study, we assumed that the deep learning model could solve the classification problem even if the raw acceleration data have noise. As a preprocessing step, we divided the data into time series using a sliding window method, and we did not conduct further preprocessing.

### 4.3. Experiment Conducted to Identify Important Frequency

Figure 3 shows the results of the experiment conducted to determine the important frequency band for each activity. Figure 3a shows that the accuracy of stay did not change after the experiment, suggesting that the DC component at 0 Hz was important. Figure 3c shows that the accuracy increased when the frequency around 1 Hz was masked. Figure 3e shows that the recognition accuracy decreased when the frequency around 1 Hz was masked. This frequency was important for stUp. Masking presumably improved accuracy because it enabled the correct classification of jog data that had been misclassified as stUp. Table 1 shows the important frequency for each activity. Relatively slow-moving activities, such as walk and stUp, had low important frequency, while relatively fast-moving activities, such as jog and stDown, had high important frequency. Based on these results, we created a filter that emphasized the frequency around the selected frequency.

### 4.4. Ablation Study

#### 4.4.1. Experimental Procedure

In the training emphasis phase, six models were trained since HASC has six different activities. In ablation study, we compared eight models listed in Table 2 to evaluate the effectiveness of DA, TTA, and EL proposed in this paper. (a) is our proposed method. (b) uses DA and TTA with a single model. (c) uses DA with EL in which each branch uses the same original sensor value in testing phase. (d) uses only DA with a single model. (e) uses TTA with EL in which each branch is trained using the same original sensor value. (f) only uses TTA. (g) is a simple ensemble learning, and (h) is a simple single model. In (b) and (f), a single model makes six predictions, applying a different enhancement filter to the test data $x_{test}$ each time a prediction is made.

#### 4.4.2. Results

Table 2 shows the validation results: the highest accuracy in bold and underlined and the second-highest accuracy in underlined. Our proposed method (a) comprised of the ensemble learning method with the frequency band enhancement filter applied during training and testing had the highest accuracy, demonstrating the effectiveness of the proposed method. Comparing (a) with (b), (c), and (e), the difference between (b) and (a) was the largest. This suggests that ensemble learning contributed the most to the improvement in accuracy. Comparing (a) and (g), the accuracy of (a) was about 1% higher than that of (g), suggesting that applying a frequency band enhancement filter to the dataset was effective. Between (f) and (h), (f) had lower accuracy. This may be because an enhancement filter was used only during testing, and data that could not be classified by the features learned during training were inputted, resulting in low accuracy. Between (b) and (h), (b) had higher accuracy, suggesting that the use of the enhancement filter during testing was more effective when combined with its use during training.

### 4.5. Effects of Window Functions

Table 3 shows the validation results: the highest accuracy in bold and underlined and the second-highest accuracy underlined. Accuracy was highest when using the Gaussian window, suggesting that a Gaussian window is appropriate for creating a frequency band enhancement filter. The lowest accuracy was obtained when a random filter was applied, suggesting that an emphasis on the important frequency band of each activity when creating a filter contributes to higher activity recognition accuracy. Furthermore, recognition accuracy was lower when the peak window was applied than when the Gaussian window was applied, suggesting that it is more effective to emphasize the frequency around the important frequency than a single frequency.

### 4.6. Validation Using Multiple Datasets

#### 4.6.1. Datasets

To evaluate the robustness of the proposed method, we conducted experiments comparing some public datasets: HASC, UniMiB [30], PAMAP2 [31], and HHAR [32]. In this experiment, we adopted “VGG16” as a single baseline model, “Ensemble learning” as a simple ensemble model, and “Proposed method” combining DA using Gaussian window, TTA, and EL. In UniMiB, we randomly sampled the acceleration data from 20 persons for training, five for validation, and another five for testing. The window size was 151 samples, divided into time series. There were 17 activities in total. In PAMAP2, we randomly sampled the acceleration data from five persons for training, two for validation, and another two for testing. The window size was 256 samples, and the stride size was 128 samples for time series segmentation. There were 12 activities in total. In HHAR, we randomly sampled the acceleration data from five persons for training, two for validation, and another two for testing. The window size was 256 samples, and the stride size was 256 samples for time series partitioning. There were six different activities.

#### 4.6.2. Results

Figure 4, Figure 5 and Figure 6 show the results of phase 1 of the proposed method to investigate the important frequency of different activities in UniMiB, PAMAP2, and HHAR, respectively. Table 4 shows the accuracy of the three models using each dataset. The highest accuracy for each dataset is shown in bold. The proposed method had higher accuracy than the ensemble learning when using HASC, PAMAP2, and HHAR and lower accuracy than the ensemble learning when using UniMiB. Thus, the effectiveness of the proposed method was demonstrated in three of the four datasets. This indicates that the proposed method is robust in different domains. Table 4 shows that the difference in accuracy between ensemble learning and the proposed method is smaller than the difference in accuracy between VGG16 and ensemble learning. This suggests that the effect of the improvement in accuracy by ensemble learning is greater than the application of the frequency-enhancement filter. In PAMAP2 the accuracy of the proposed method is a little less than the conventional ensemble method but reaches almost the same estimation accuracy. The proposed method employs a frequency-enhanced method for each activity label compared to the conventional ensemble. Therefore, the proposed method may not be more effective than the conventional ensemble method when there are a very large number of behaviors and when similar behaviors are included. It can also be seen that the accuracy of ensemble learning is higher than the accuracy of the proposed method when the UniMiB dataset is used. This may be due to the fact that, as shown in (b), (c), and (p) in Figure 6, there are more activities with a smaller decrease in recognition accuracy when mask processing is performed than in the other datasets.

Table 5 shows that the important frequencies for Falling Right and Falling Left are identical. This is thought to be because they are almost identical activities, differing only in the direction of falling. Table 6 shows that, similar to HASC, PAMAP2 was less important for relatively slow-moving activities such as lying and sitting and more important for relatively fast-moving activities such as running and rope jumping. Table 7 shows that the HHAR includes only relatively slow-moving activities, which may account for the lower important frequencies.

## 5. Conclusions

In this study, in order to improve the accuracy of activity recognition prediction and to develop a variety of activity recognition services, we proposed a general-purpose method based on frequency enhancement and ensembles. The proposed method (1) finds important frequency in predicting each activity and creates a filter that emphasizes the found frequency, (2) trains the model by applying the filter to training data, and (3) performs ensemble learning by applying the filter to testing data. 

The experiments conducted to identify the important frequency of each activity revealed that the DC component of stay (0 Hz) was important. Relatively slow-moving activities are expected to have a lower important frequency, while relatively fast-moving activities are expected to have a higher important frequency.

Ablation study results showed that the proposed method combining emphasis during training and testing and ensemble learning resulted in the highest recognition accuracy. Ensemble learning was the element that contributed the most to the accuracy of the proposed method. The frequency band enhancement filter was effective when applied to both the training and testing data but not when applied only to the testing data.

In an experiment conducted to examine the effect of the window function, four different filtering patterns were tested and compared in terms of recognition accuracy. Accuracy was highest when the filter was created with a Gaussian window and lowest when a random filter was applied, suggesting that emphasizing important frequency when creating filter results in higher accuracy. In addition, although this study proposes a method of emphasizing important frequencies for each activity, it is thought that the accuracy of recognition may be further improved by emphasizing or weakening the frequencies according to their importance.

An experiment was conducted to verify the robustness of the proposed method in different domains. The results showed that the proposed method performed better than an ensemble learning method in three out of four datasets (HASC, PAMAP2, and HHAR), demonstrating its robustness in different domains.

In this study, we used VGG16 in the phase of finding important frequencies and would like to experiment to see if the important frequencies change depending on the structure of the model. Additionally, the most important frequency of each activity is emphasized to improve the estimation accuracy of activity recognition. In addition to emphasizing the most important frequencies, we believe that the recognition accuracy can be further improved by emphasizing or de-emphasizing the frequencies according to their importance. We would like to create a frequency band enhancement filter other than the one used in this study and verify the change in accuracy. In addition, since the range of values applied to the amplitude spectrum in this study was between 0.5 and 1, we would like to investigate how the accuracy changes when the values are varied. We would also like to further improve the accuracy by using deep learning to create the frequency filter itself. As described above, we believe that the recognition accuracy can be improved over the current accuracy by changing the method of creating the frequency band enhancement filter.

## Figures and Tables

**Figure 1 sensors-23-01465-f001:**
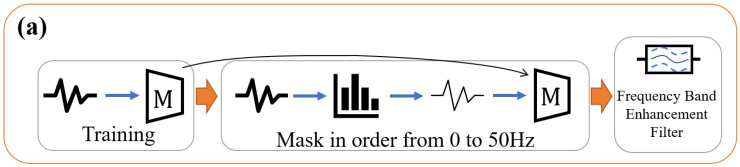
Overview of the proposed method. (**a**) Phase 1: Finding the important frequencies of each activity; (**b**) Phase 2: Emphasis during training; (**c**) Phase 3: Emphasis during testing.

**Figure 2 sensors-23-01465-f002:**
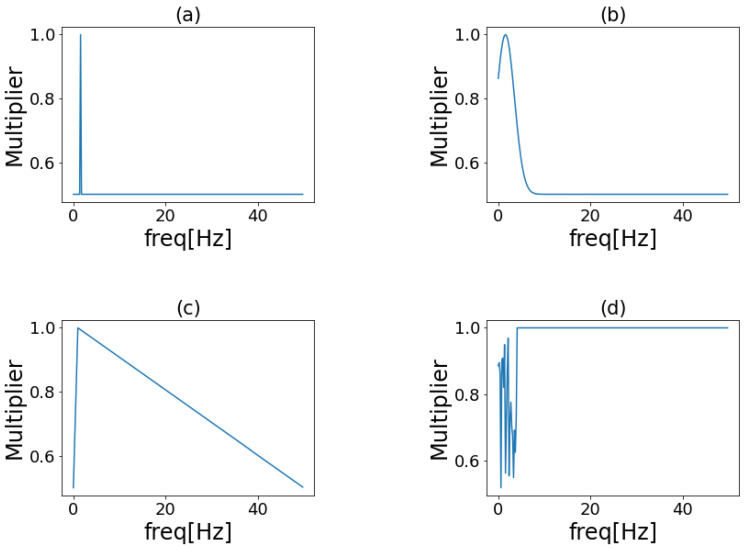
Examples of frequency band enhancement filters. (**a**) Peak window; (**b**) Gaussian window; (**c**) Triangular window; (**d**) Random window.

**Figure 3 sensors-23-01465-f003:**
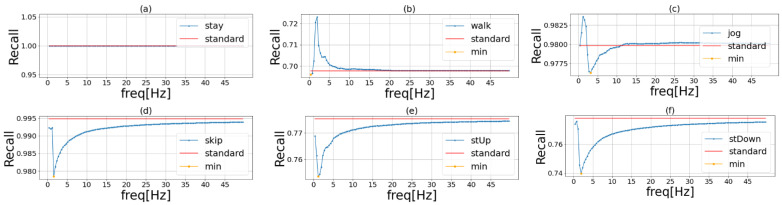
Important frequency of each activity in HASC. The blue line in the graph shows the recognition accuracy when we masked the frequencies of the original sensor data in order. The red line shows the recognition accuracy when we used the original data. The yellow point is the lowest recognition accuracy when we use the frequency masked data. Each activity is (**a**) stay, (**b**) walk, (**c**) jog, (**d**) skip, (**e**) stUp, and (**f**) stDown.

**Figure 4 sensors-23-01465-f004:**
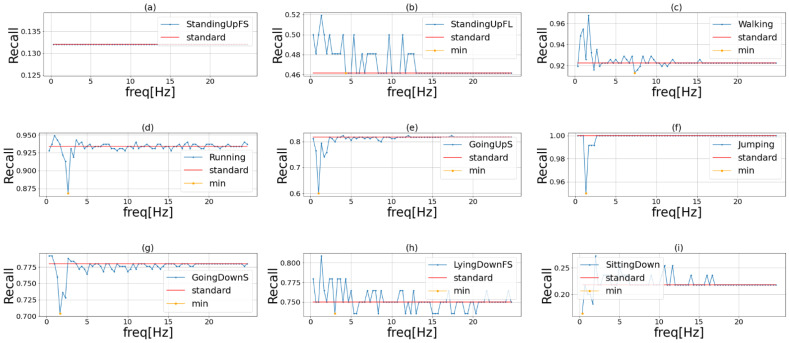
Important frequency of each activity in UniMiB. The blue line in the graph shows the recognition accuracy when we masked the frequencies of the original sensor data in order. The red line shows the recognition accuracy when we used the original data. The yellow point is the lowest recognition accuracy when we use the frequency masked data. Each activity is (**a**) StandingUpFS, (**b**) StandingUpFL, (**c**) Walking, (**d**) Running, (**e**) GoingUpS, (**f**) Jumping, (**g**) GoingDownS, (**h**) LyingDownFS, (**i**) SittingDown, (**j**) FallingForw, (**k**) FallingRight, (**l**) FallingBack, (**m**) HittingObstacle, (**n**) FallingWithPS, (**o**) FallingBackSC, (**p**) Syncope, (**q**) FallingLeft.

**Figure 5 sensors-23-01465-f005:**
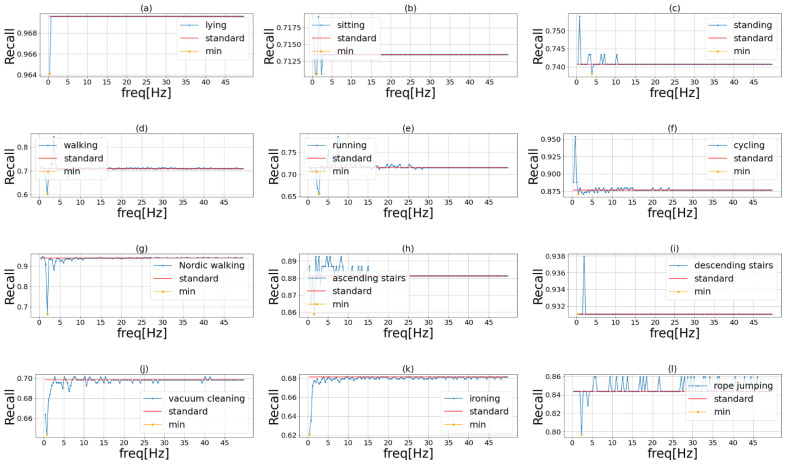
Important frequency of each activity in PAMAP2. The blue line in the graph shows the recognition accuracy when we masked the frequencies of the original sensor data in order. The red line shows the recognition accuracy when we used the original data. The yellow point is the lowest recognition accuracy when we used the frequency masked data. Each of these activities is (**a**) lying, (**b**) sitting, (**c**) standing, (**d**) walking, (**e**) running, (**f**) cycling, (**g**) Nordic walking, (**h**) ascending stairs, (**i**) descending stairs, (**j**) vacuum cleaning, (**k**) ironing, and (**l**) rope jumping.

**Figure 6 sensors-23-01465-f006:**
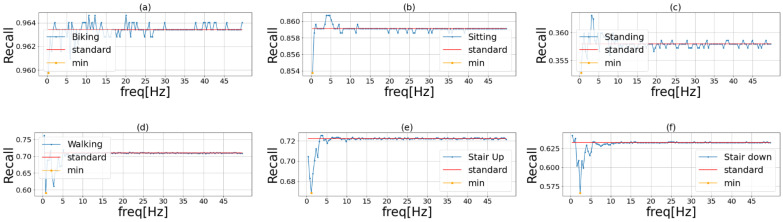
Important frequency of each activity in HHAR. The blue line in the graph shows the recognition accuracy when we masked the frequencies of the original sensor data in order. The red line shows the recognition accuracy when we used the original data. The yellow point is the lowest recognition accuracy when we used the frequency masked data. Each activity is (**a**) Biking, (**b**) Sitting, (**c**) Standing, (**d**) Walking, (**e**) Stair Up, and (**f**) Stair down.

**Table 1 sensors-23-01465-t001:** Important frequency of each activity in HASC.

Activity	Frequency (Hz)
stay	0.00
walk	0.78
jog	3.13
skip	1.56
stUp	1.17
stDown	1.95

**Table 2 sensors-23-01465-t002:** Accuracy of the eight methods used in the experiment. DA, TTA, and EL are denoted by ✓ for those applied and x for those not applied. The highest accuracy is underlined and bolded, and the second highest is underlined.

Method	DA	TTA	EL	Accuracy
(a)	✓	✓	✓	** 0.890 **
(b)	✓	✓	x	0.877
(c)	✓	x	✓	0.881
(d)	✓	x	x	0.876
(e)	x	✓	✓	0.880
(f)	x	✓	x	0.845
(g)	x	x	✓	0.880
(h)	x	x	x	0.873

**Table 3 sensors-23-01465-t003:** Accuracy when each filter was applied. The highest accuracy is underlined and bolded, the second highest is underlined.

Filter Type	Accuracy
Peak window	0.890
Gaussian window	** 0.896 **
Triangular window	0.888
Random	0.872

**Table 4 sensors-23-01465-t004:** Accuracy of the three models using each dataset. Bold type indicates the highest accuracy using the respective dataset.

Method	HASC	UniMiB	PAMAP2	HHAR
VGG16	0.873	0.663	0.787	0.715
Ensemble learning	0.880	**0.707**	0.812	0.753
Proposed method	**0.896**	0.703	**0.818**	**0.760**

**Table 5 sensors-23-01465-t005:** Important frequency of each activity in UniMiB.

Activity	Frequency (Hz)
StandingUpFS	0.00
StandingUpFL	4.30
Walking	7.28
Running	2.65
GoingUpS	0.99
Jumping	1.32
GoingDownS	1.65
LyingDownFS	2.98
SittingDown	0.33
FallingForw	1.65
FallingRight	0.66
FallingBack	0.33
HittingObstacle	3.31
FallingWithPS	0.33
FallingBackSC	0.33
Syncope	1.98
FallingLeft	0.66

**Table 6 sensors-23-01465-t006:** Important frequency of each activity in PAMAP2.

Activity	Frequency (Hz)
lying	0.39
sitting	0.78
standing	3.91
walking	1.95
running	2.73
cycling	1.56
Nordic walking	1.95
ascending stairs	1.56
descending stairs	0.39
vacuum cleaning	0.78
ironing	0.39
rope jumping	2.34

**Table 7 sensors-23-01465-t007:** Important frequency of each activity in HHAR.

Activity	Frequency (Hz)
Biking	0.39
Sitting	0.39
Standing	0.39
Walking	0.78
Stair Up	1.17
Stair Down	2.34

## Data Availability

Data sharing not applicable.

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
