# Peer review of "Sensor-Based Activity Recognition Using Frequency Band Enhancement Filters and Model Ensembles"

_sensors, 2023, doi:10.3390/s23031465_

Round 1

Reviewer 1 Report

Summary

In the article “Sensor-Based Activity Recognition Using Frequency Band Enhancement Filters and Model Ensembles”, the authors present a methodology to identify important frequencies of activities (based on accelerometer/gyroscope data), that can be used to create an enhancement filter, and further used to improve the classification accuracy of any classifier. The manuscript looks at multiple datasets, to justify the improvements in classification accuracy based on the proposed methodology.

Comments for Authors

1.     More information should be provided in the Introduction section particularly about the wider impact of this research, and significance of the problem addressed by this manuscript (along with appropriate references).

2.     Is the CNN model used in Phase 1 (identification of important frequency for each activity) a generic CNN model? Why is a CNN model chosen here, and can it be replaced with other classification algorithms?

3.     Based on the results shared in Table 2, the proposed method (Method (a) in the table) has a classification accuracy of 89%, while the simple single model (Method (h) in the table) has a classification accuracy of 87.3%. Therefore, the improvement in classification by the proposed methodology is only ~2%. The authors should discuss how this can be further improved and what are the limitations of their proposed method.

Reviewer 2 Report

The manuscript proposes an activity recognition method using ensemble learning and filters. It emphasizes the critical frequency for recognizing a specific activity and then creates a filter to enhance the identified frequency band. The presented method and the achieved results are quite interesting; however, some concerns need to be clarified.

  1.  Three different methods, including the proposed method, were used in this research. The reviewer suggests adding a sentence to the abstract and mentioning how much the proposed method improved the results compared to the other two.
  2. In section 4.1, the authors briefly explained the applied experiment. Did the used dataset consider the effect of noise and interference? How can the possible noise affect the accuracy of the recognition? Please clarify.
  3. In Table 4, the highest accuracy was achieved using the proposed method for all datasets except UniMiB. Do the authors think this could be due to the different window size samples? Please explain.
  4. The list of references may be revised by adding more recently published journal articles.

Reviewer 3 Report

1. Author needs to improve the introduction part

2. design and prposed work

3. resuolts and discussions, the caption of figures must be explained well

4. the conclusion shoudl be improved

5. improve grammer of MS

6. Authors have cited only 18 refrences, they must add more recent refernces.

Round 2

Reviewer 2 Report

The current manuscript version was appropriately revised, and the authors satisfactorily replied to the reviewer's comments. The reviewer believes the manuscript can be published in the current version. However, the language and organization of the manuscript could be further improved.